# Development of Aerobic Exercise Equipment Using Universal Design: Treadmill and Arm Ergometer

**DOI:** 10.3390/healthcare10112278

**Published:** 2022-11-14

**Authors:** Eunsurk Yi, Hyun Byun, Ahra Oh

**Affiliations:** 1Department of Exercise Rehabilitation & Welfare, Gachon University, 191 Hombakmoero, Yeonsu-gu, Incheon 406-799, Korea; 2Exercise Rehabilitation Convergence Institute, Gachon University, 191 Hombakmoero, Yeonsu-gu, Incheon 406-799, Korea

**Keywords:** universal design, exercise participation of the disabled, aerobic exercise equipment, treadmill, arm ergometer

## Abstract

Exercise products based on universal design, which reduce restrictions on the exercise environment and ensure convenience and safe use, are beneficial for people with a disability; however, the current universal design only considers the preferences of the general population, which is not suitable for the disabled population. This results in the exclusivity of the sports facilities and supplies for people with a disability. Consequently, we explored the components of universal design and product satisfaction by considering users with disabilities and proposed the direction for designing extended universal exercise equipment that is suitable for them. Specifically, this study focuses on developing exercise equipment for people with a disability. Based on the results from the evaluation of acceptance and satisfaction of universal sports products for people with a disability using design thinking, we suggest the following. First, it is necessary to consider safety devices for exercise products. Second, the user interface should be improved in terms of convenience. Third, the ergonomic instrument design should be improved. Finally, the instrument design should be centered on user convenience.

## 1. Introduction

Disability hinders the ability to perform daily life activities depending on the level or type of the disability. According to the World Bank report in 2022, 15% of the world’s population has a disability, and Korea accounts for 5% of this total population [1]. The high proportion of people with a disability compared to the total population illuminates the various needs of the disabled population in society. Furthermore, the participation rate of people with a disability in daily sports is rapidly increasing; consequently, the demand for new welfare and sports equipment for this group is increasing [2].

The Ministry of Culture, Sports, and Tourism of Korea reported that the rate of participation in exercise by people with a disability tripled from 8.6% in 2010 to 24.9% in 2019 [3]. This indicates that the interest of people with a disability in sports activities has increased significantly. In addition, the more experienced people with a disability become, the higher the demand for support equipment during exercise; therefore, there is an increasing need to develop exercise equipment that all categories of people, including those with a disability, can use [3]. Compared to the normal population, people with a disability are at a disadvantage in terms of educational, socioeconomic, and occupational opportunities [4,5]. Moreover, the regular exclusion of people with a disability from design, service, and policy-making makes them less active and results in their isolation, depression, and loneliness [6].

Irregular physical activity among people with a disability may lead to adult and inactivity-related diseases [7]. According to the Center for Disease Control and Prevention (CDC), adults with disabilities are more likely to stop participating in physical activities in their leisure time than those with no disability. In addition, among teenagers, girls with disabilities in grades 9–12 reported at least 60 min of physical activity per day, for 3.1 days per week, whereas girls with no disabilities reported the same for 4.5 days per week [8]. The decrease in exercise participation among people with a disability reduces health-related benefits. In addition, the lack of regular physical activity can negatively affect their psychological and social health. This is because it can deepen their feeling of isolation, increase stigma, and worsen their social relations [9].

Therefore, gymnasiums and fitness centers for people with a disability are preferred places for improving their quality of life by participating in physical activities [10,11]. Fitness centers appeal to a wide range of disabled sports participants because they are relatively easily accessible, have flexible operating hours, and have no requirements for specific physical skills or fitness levels. However, the nature of current sports facilities still restricts people with disabilities from participating in sports due to their design, which does not consider the peculiarities of these groups. When efforts are required to use sports equipment, time and difficulty in procuring them can increase the stress of people with a disability [6]. For example, if these facilities lack user-friendly features for people with a disability, such as easy access to wheelchairs, which greatly undermines the emotional well-being of these participants [12]. Therefore, the design of sports facilities and equipment/devices should be user-friendly for people with a disability and non-disability in a diverse society. This means that more emphasis is needed on the application of universal design (UD).

UD was introduced in the 1970s by Ronald Mace, an American architect and product designer with a disability [13,14]. Ronald defined UD as a “product and environmental design that can be used by as many people as possible without adaptive or specialized design”. Between 1994 and 1997, the National Institute on Disability and Rehabilitation Research (NIDRR), which is responsible for disability and rehabilitation among U.S. government agencies, made a major investment in the study of universal design. In the process, the Center for Universal Design introduced the following seven UD principles for architects, product designers, engineers, and environmental design researchers: (1) Equitable use, (2) flexible use, (3) simple and intuitive use, (4) perceptible information, (5) tolerance for error, (6) low physical effort, and (7) size and space for approach and use. Reportedly, various product designers have modified their products to accommodate these seven principles to increase usability by both people with a disability and those with no disability [15].

Exercise equipment facilitates educational and vocational social rehabilitation for people with a disability by mitigating physical and functional disabilities. Moreover, exercise equipment for people with a disability is an essential element that enables daily life and social activities and is a necessary means of improving the quality of life for the disabled and expanding opportunities for integration and independence. Equipment for convenience improvement and recreational activities for people with a disability has a relatively short history. For example, lightweight folding wheelchairs, which were first developed in the United States in 1933 [16], are still not very popular. Rimmer et al. [17] found significant opportunities to enable people with a disability to exercise and play in fitness centers and swimming pools that comply with the Accessibilities Disabilities Association (ADA) standard. In addition, Ludwa and Lieberman [18] studied the application of UD with the intention of enabling students with disabilities to participate in spikeball, which is a new sport.

Various studies in the field of sports have examined the applicability of UD to products for people with a disability. Gray, Zimmerman & Rimmer [19] conducted a study to measure the level at which the UD of human-made structures guarantees the physical activity of the disabled. Additionally, Kim & Chang [20] measured the user’s perception of sports facilities by applying universal design. Currently, UD products are being actively researched around the world. In Korea, the exercise participation rate of people with a disability is reported to be low due to the lack of exercise infrastructure that is suitable for this group. The Ministry of Culture, Sports, and Tourism [3] surveyed sports facilities mainly used by the people with disability. The survey showed that 8.1% of public sports facilities, 3.9% of private sports facilities, and 3% of public facilities for people with a disability, and it was found that 81.5% of people with a disability did not use these sports facilities. Furthermore, 37.7% of people with a disability were found to have difficulty exercising alone because they did not use sports facilities. The Korean government has decided to build 150 new sports centers for people with a disability by 2025 to increase their participation in daily sports. The Sports Center for the Disabled, called the Bandabi Sports Center, will be operated as an integrated facility where people with a disability will have the right to use it first, while those with no disability will also use it. This implies that the need to develop exercise products with universal design, which enables people with a disability and those with no disability to participate in the exercise, is gradually increasing. Therefore, the purpose of this study is to develop an aerobic exercise equipment design, which employs a UD that can be used by people with a disability, those with no disability, and the elderly by applying the design thinking method.

## 2. Methods

### 2.1. Procedure

This study is based on the principle of “design thinking”. Self-thinking-based research is research that seeks a problem-solving process centered on people, and it is characterized by providing the process for the optimal combination of problem-solving, business, intuition and logic, etc. [21,22]. Therefore, to design universal exercise equipment, we conducted this study by applying the “design thinking” principle. Although the process of “Design Thinking” is diverse, the “Double-Diamond” process is presented among them, which consists of four steps—“Discover”, “Define”, “Develop”, and “Deliver” [23]. Figure 1 explains the double diamond design process.

The first step is the discovery step, where ideas are reviewed, reclassified, and selected. In other words, it refers to literature and field research processes such as market research and user research. The second stage is the define stage, where the direction is established. The third stage is the development process, where problem-solving ideas are proposed through various methodologies. The final stage is the delivery stage, which refers to the process of deriving and testing the final concept through the process of converging the diffusion of various ideas and applying the design [23]. The overall procedure of the study is shown in Figure 2.

### 2.2. Design Development

#### 2.2.1. Discover

In this stage, a basic investigation of UD sports equipment was carried out. Market research on sports equipment was conducted, and a focus group interview (FGI) was conducted with trainers at 13 sports facilities used by both disabled and non-disabled people. Through the basic survey, the arrangement status of exercise equipment, problems, and demands for exercise equipment was investigated and analyzed. The characteristics of the participation of the FGI are shown in Table 1.

For FGI, data were analyzed according to the FGI content analysis method proposed by Krueger and Casey [24]. First, to rule out study bias, the FGI contents were transcribed in the form of oral transcripts by researchers not subordinated to the study. Second, the transcribed contents were compared with the audio file again, and the observation results were added to the transcript based on the chapter memos developed during the interview process. Third, the data was transcribed into Microsoft^®^ Office Word 2019 (Microsoft Corporation, Redmond, WA, USA) and read iteratively to analyze the content of the raw data. Then, meaningful words were entered into Microsoft^®^ Office Excel 2019 (Microsoft Corporation, Redmond, WA, USA) to extract the theme. We grouped the common features of the subjects and then reclassified keyword. By repeating the classification process, we were able to identify and analyze common attributes in the responses in terms of frequency, specificity, emotion, and abundance [25]. The characteristics of the participation of the FGI are shown in Table 1.

#### 2.2.2. Define

Based on the content analyzed in the “Discover” stage, concept ideas were presented according to the improvement elements and requirements of the existing exercise equipment through an expert meeting, and the components of each product were identified to classify the ideas. Improvement plans for problems that may occur in the process of using the product concept ideas were derived through the first design concept by classifying ideas into items of usability, safety, economic feasibility, sustainability, and aesthetics according to the major UD elements for each derived idea.

Experts conducted semi-structured interviews with the participants. Interviews were conducted until data was saturated, coded and coded in Microsoft^®^ Office Excel 2019 (Microsoft Corporation, Redmond, WA, USA), and analyzed thematically through repeated perusal. The validity of the study was determined using the method proposed by Guba and Lincoln [26]. The final results were shared in writing or emailed to 10 experts for review and feedback on what needs to be corrected or deleted. The results confirmed the accuracy of the final study results. The participants of the expert meeting are shown in Table 2.

#### 2.2.3. Develop

The acceptance of the aerobic exercise equipment design concept idea to which the first UD derived from the ‘Define’ stage was applied was evaluated. A total of 396 users (131 seniors, 133 disabled people, and 132 non-disabled people) from 13 sports facilities used by people with a disability and those with no disability were selected as the participants. To determine the acceptability, frequency analysis was performed using IBM SPSS Statistics 23 (IBM, Armonk, NY, USA). The characteristics of the first design concept idea acceptance evaluation participants are shown in Table 3.

From the evaluation of acceptance of the first design concept idea, two types of 3D models of the treadmill and arm ergometer, which are aerobic exercise equipment based on universal design, were developed. Furthermore, these models were reviewed by experts. Afterward, the design satisfaction was evaluated using 60 users (including both people with a disability and those with no disability) from sports facilities. The satisfaction evaluation was determined by using IBM SPSS Statistics 23 (IBM, Armonk, NY, USA) to conduct frequency analysis. The characteristics of 3D design satisfaction evaluation participants are shown in Table 4.

#### 2.2.4. Deliver

A secondary design concept was prepared based on the design satisfaction evaluation and 3D modeling design concept idea acceptance evaluation results derived through the ‘Develop’ stage, and the final design was created by modifying and supplementing the design concept through the expert meeting. The analysis method of the results of the expert meeting was the same as the analysis method performed in the “Define” step.

## 3. Results

### 3.1. Discover

#### 3.1.1. Exercise Equipment Market Research Results

Market research was conducted on exercise equipment that can be used by both people with a disability and those with no disability. The market research results are shown in Table 5.

#### 3.1.2. Analysis of the Status of Aerobic Exercise Equipment in Sports Facilities Used by Both Disabled and Non-Disabled People

A focus group interview was conducted on the current status of trainers and aerobic exercise equipment at 13 public sports facilities used by people with a disability and those with no disability together, and the related problems and requirements were also investigated. Table 6 shows the results of the FGI.

### 3.2. Define

#### 3.2.1. Deriving Concept Ideas

In the ‘Discover’ stage, the components of each product of the existing aerobic exercise equipment researched in the market were identified, the ideas were classified, and the FGI results were reflected to derive the concept ideas according to the improvement factors and the user’s needs.

To derive the conceptual idea of treadmill, first, the main characteristics and operation methods of treadmill investigated in the existing market were analyzed. Second, through FGI, the concept idea of a treadmill applied with a universal design was developed by reflecting the problems of the existing treadmill and the needs of users. The concept idea of the derived treadmill is shown in Figure 3 and Table 7.

To derive the conceptual idea of an arm ergometer, the main characteristics and operation methods of the arm ergometer investigated in the existing market were primarily analyzed. Second, through FGI, the concept idea of an arm ergometer applied with a universal design was developed by reflecting the problems of the existing arm ergometer and the needs of users. The concept idea of the derived arm ergometer is shown in Figure 4 and Table 8.

#### 3.2.2. Deriving the First Design Concept Idea

The design concept idea for aerobic exercise equipment was developed from the first design concept idea by classifying ideas into usability, stability, economic feasibility, sustainability, and aesthetics items according to UD elements. The design concept ideas were derived from six types of treadmills with UD and six types of arm ergometers with universal design. The first design concept idea is shown in Figure 5 and Figure 6.

### 3.3. Develop

#### 3.3.1. Concept Idea Acceptance Evaluation in Progress

The acceptance of the first design concept idea was evaluated for 396 users from sports facilities used by both disabled and non-disabled people. The concept ideas were designed for six treadmills and six-arms ergometers, which were based on universal design, and the acceptance of each concept was evaluated. The evaluation results for each concept idea applying UD are shown in Figure 7 and Figure 8.

#### 3.3.2. Design Development of Aerobic Exercise Equipment Based on Universal Design

Among the aerobic exercise products, which were applied with universal design, six treadmills and six arm ergometers were found to have accepted the concept ideas. The acceptance results were reviewed by experts and the ideas were classified into usability, stability, economic feasibility, sustainability, and aesthetics items according to UD elements through a meeting. Based on this content, 3D models of two treadmills and two-arm ergometers were created and reviewed by experts. The first 3D modeling is shown in Figure 9 and Figure 10.

#### 3.3.3. Product Design Satisfaction Evaluation

Product Design Satisfaction Evaluation is based on 3D modeling based on two types of treadmill and ergometers, which are aerobic instruments that apply universal designs to 60 people who participate in an exercise in the gym used by the disabled and non-disabled people. A design satisfaction evaluation was conducted. Satisfaction evaluation evaluated the attribute evaluation and preference type of the product.

As a result of the evaluation of treadmill properties, the evaluation of ‘type B’ was high in most of the properties. In addition, it was found that the participants with disabilities had a particularly high evaluation of the ‘safety of use’ attribute. The result of the attribute evaluation of treadmill product design is shown in Figure 11.

The preferred treadmill design was type B (63.3%). The reason for the preference for type B was ‘safety of use’ (71.1%), followed by ‘convenience of use’ (57.9%) and durability/strength (7.9%). The evaluation result of the preferred type of treadmill product design is shown in Figure 12.

As for the properties of the arm ergometer, the evaluation of type B was high. Participants with disabilities were found to have a particularly high evaluation of the ‘safety of use’ attribute. The result of the attribute evaluation of the arm ergometer product design is shown in Figure 13.

As for the preferred arm ergometer design, type B (78.3%) was overwhelmingly high. As the reason for the preference for type B, ‘convenience of use’ was found to be the highest, followed by ‘visual beauty’ (30.8%) and ‘durability/strength’ (19.1%). The evaluation result of the preferred type of arm ergometer product design is shown in Figure 14.

### 3.4. Deliver

#### 3.4.1. Design Acceptability Verification

Based on the acceptance and satisfaction evaluation results of the first aerobic exercise equipment applying UD, which was first developed in the develop stage, the design concept was prepared by modifying the product design.

The consideration for applying the UD element to the existing treadmill consisted of a running belt, auxiliary footrest, handle and auxiliary handrail, display, operation button, and treadmill installation method. The second concept design and guideline of the treadmill to which UD is applied are shown in Figure 15 and Table 9.

To apply the UD element to the existing arm ergometer, the saddle, handle, case, display, information board, and installation environment must be considered. The second concept design and guidelines for the UD-applied arm ergometer are shown in Figure 16 and Table 10.

#### 3.4.2. Design Completion

The design concept was revised and supplemented through expert evaluation. Based on the main considerations in the design of each product, the final product design was made by referring to the detailed example images based on the guidelines.

The main features of the final 3D design of the treadmill with UD are shown in Figure 17 and Figure 18.

The main features of the final 3D design of the arm ergometer with UD are shown in Figure 17.

## 4. Discussion

Using the design thinking process, this study explored five stages of the universal design development process. First, this study examined the demands of sports equipment for people with a disability. Second, it extracted design elements. Third, it evaluated the acceptance and satisfaction with the design by people with a disability. Fourth, an expert evaluation was conducted. Bianco [27] stated that people with physical and cognitive disabilities need accessible, usable, and safe designs. Currently, commercialized exercise devices provide insufficient control over safety risks for people with disabilities. When people with a disability use exercise equipment, operational support is needed to control various safety risks, such as the risk of falling while moving or exercising, impact from the use of equipment, and injuries caused by inexperience in operation [28]. Specifically, when a user with a disability performs upper and lower body exercises using an exercise device, the user’s weight or driving torque should be considered. In particular, in the case of people with partial paralysis, the left and right muscle strength are different; therefore, it should be designed not to tilt to one side and fall when using the instrument. In addition, the design of exercise equipment should be made in consideration of the body size of the disabled user. In other words, the user should be able to exercise in an optimal posture according to the size of the body, such as the height of the disabled patient and the length of the limbs. This is because if the posture of the disabled user is unstable, it may cause a fracture risk depending on the user. Lezzoni et al. [29] reported that patients fear injury from movement without ancillary equipment. A safety-conscious UD can reduce their fear of injury and enhance their intention to participate in the exercise. Accordingly, many researchers have researched the safety of equipment for the disabled. Agaronnik et al. [30] studied the design of diagnostic equipment for the safety of disabled patients, while Hollis et al. [31] reported increased physical activity in accessible parking spaces with paved sidewalks and curb slopes. Calder et al. [32] also reported that the fitness center’s ramp, the height of the device, and the location of the shower room device are designs that threaten the safety of the disabled. Accordingly, activating injury-preventing emotional engineering designs such as active injury-preventing systems, fall-preventing auxiliary handles, and shock-relieving materials is necessary.

Second, the user interface should be improved in terms of convenience. Specifically, for disabled people, it is difficult to check exercise information while using the exercise equipment. Accordingly, voice and visual feedback for confirming exercise information should be provided. Considering the visual and tactile interfaces attached to the exercise equipment is essential. Specifically, if the user is disabled, the user’s hands and feet should be able to be fixed to the exercise equipment. In the case of people with hemiplegia disabilities, the limbs of the uncomfortable body part are not strong enough to hold on to the handle of the exercise equipment and to keep the lower body in a fixed position, so additional devices such as straps are needed. In addition, the accessibility of wheelchairs for wheelchair users should be considered. Many previous studies have been studied on the interface of universal design. Accordingly, Arbour-Nicitopoulos and Ginis [33] argued that exercise equipment should be able to secure a wheelchair space and that a system should be equipped to monitor the physical environment. Roger-Shaw et al. [34] stated that UD should provide various means for users to acquire information and knowledge. Baida and Ivanova [35] suggested information provision according to the importance and degree of disability as a guideline. Based on various previous studies, UD for the disabled should be a design that considers auxiliary equipment and accessibility for stable use of equipment, and intuitive information delivery is an essential factor. Convenience may be improved by applying more noticeable colors to areas that require attention during exercise or by marking Braille. It also increases the user’s interest in the exercise and lacks providing exercise scenarios for each disability characteristic. The fun factor of exercise devices is a significant factor that affects exercise commitment and continuity. Therefore, building an interface that provides visual and auditory feedback is necessary.

Third, the ergonomic instrument design should be improved. In the case of wheelchair users as well as general users, equipment design should be made so that exercise can be performed at an appropriate location during its usage. Additionally, there is a need for a user-friendly design so that the operation can be performed easily and quickly when using the device. Finally, the instrument design should be centered on user convenience. In the case of handles or controls of exercise equipment, the degree of perception of discomfort varies depending on the type of disability or age, so it is necessary to consider a design that can meet their needs accordingly.

Moreover, these versatile designs are very cost effective and can benefit people with a disability and the general public. Social-ecological design for fitness facilities improves access to fitness facilities and enables proper investment recovery [36]. The total research and development cost of this study was about USD 315,000, and USD 52,000 was spent per unit. This shows a relatively high development cost compared to the development cost of general fitness equipment. However, the optimum design of a product, which considers both people with a disability and those with no disability, can ultimately maximize performance and minimize costs [37]. Applying universal design increases the accessibility of sports facilities by people with a disability, and the resistance of those with no disability to unfamiliar machines is low. Accordingly, UD will be able to fill the gap between sports facilities for people with a disability and general facilities.

## 5. Conclusions

This study focuses on designing exercise equipment for the disabled, which recently has been revealed to induce exercise participation in the disabled. To this end, we explored the components of UD and product satisfaction by considering users with disabilities and proposed the direction for designing extended universal disabled exercise equipment. Exercise products based on universal design, which reduce restrictions on the exercise environment and ensure convenience and safe use, are beneficial for groups with disabilities; however, the UD that only considers the preferences of the general population is unsuitable for the disabled population. This results in the exclusivity of the sports facilities and supplies for the disabled. Universal sports equipment for the disabled is a strategic industry that can lead the 21st century, and long-term investors should be supported. Moreover, the development of working equipment for the disabled is also a crucial task. Universal sports organizations for the disabled will provide a social environment that can improve participation in non-discriminatory sports and provide a foundation for overcoming disabilities and supporting daily social life. This implies that UD for disabilities has emerged as a novel research topic that requires more collaborations across disciplines as well as fill the research gap in the extant literature on UD to better comprehend the sustainable participation of disabilities in exercise Finally, the design of UD aerobic exercise equipment should be produced by being open to exercise equipment manufacturers. It does not end with design development research; however, the value of this design development research will increase when the design is utilized for mass production by manufacturers and when the product is used by people.

## Figures and Tables

**Figure 1 healthcare-10-02278-f001:**
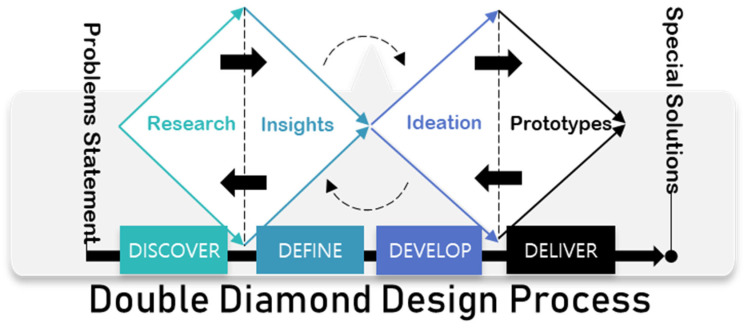
Double diamond design process.

**Figure 2 healthcare-10-02278-f002:**
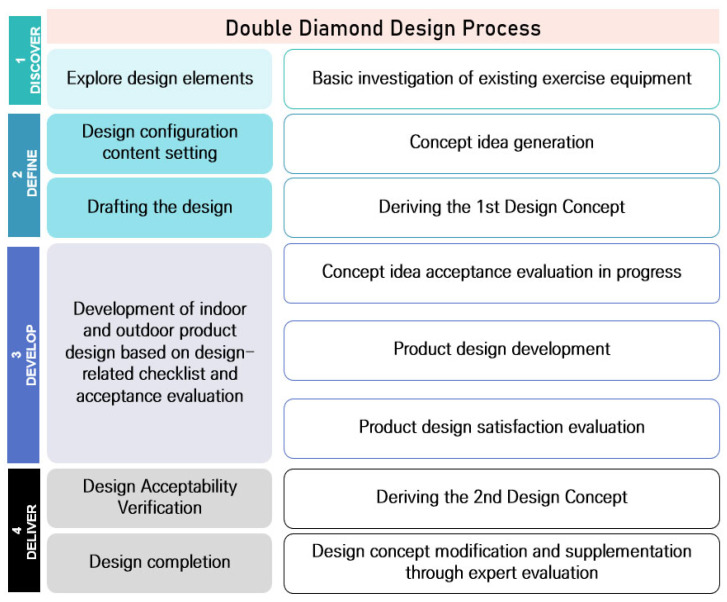
Research procedure.

**Figure 3 healthcare-10-02278-f003:**
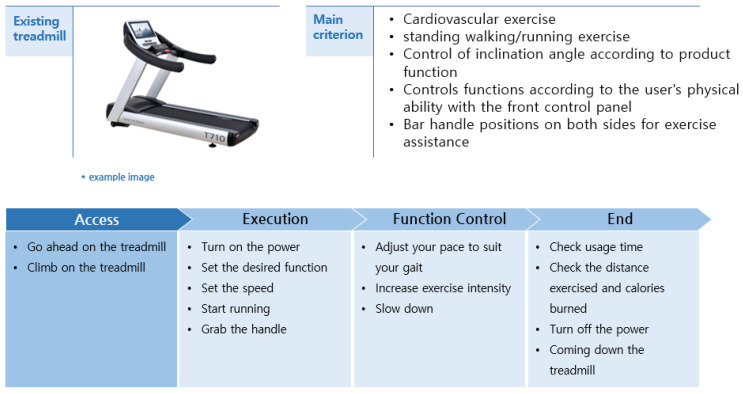
Treadmill concept idea.

**Figure 4 healthcare-10-02278-f004:**
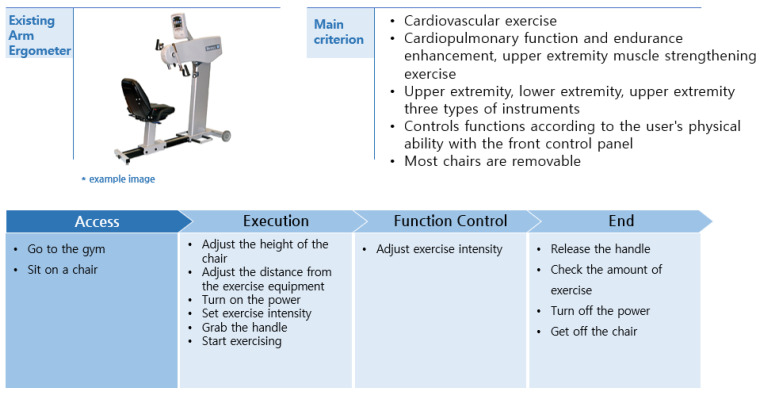
Arm ergometer concept idea.

**Figure 5 healthcare-10-02278-f005:**
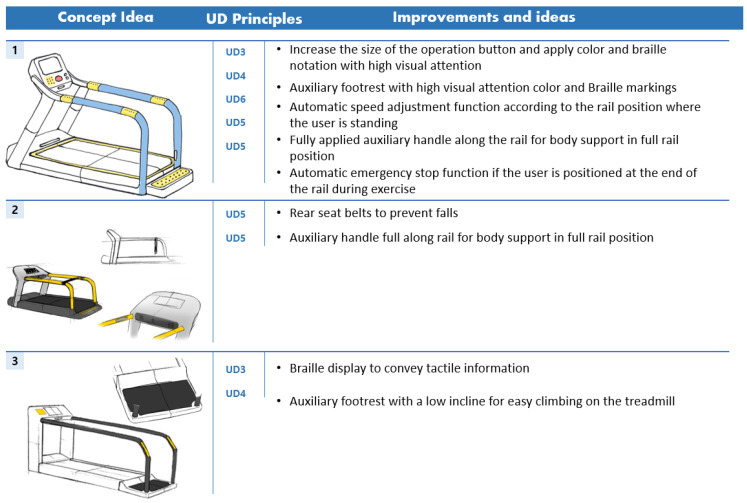
Six types of treadmill concept ideas applied with UD.

**Figure 6 healthcare-10-02278-f006:**
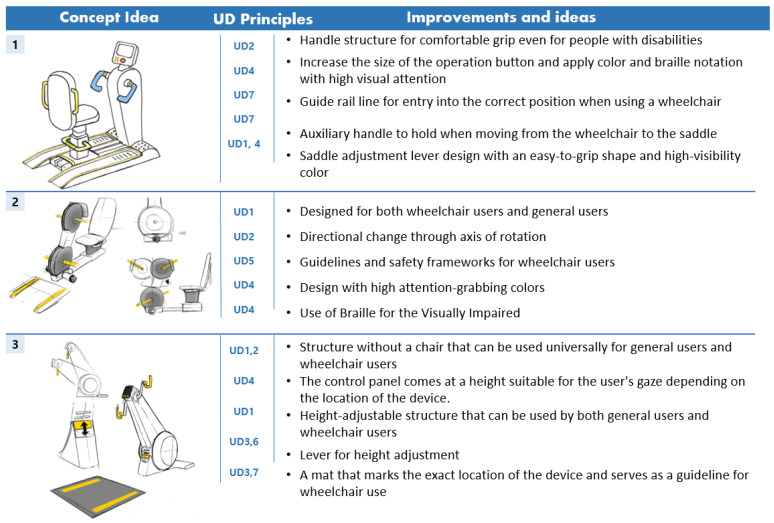
Six types of concept ideas for an arm ergometer based on UD.

**Figure 7 healthcare-10-02278-f007:**
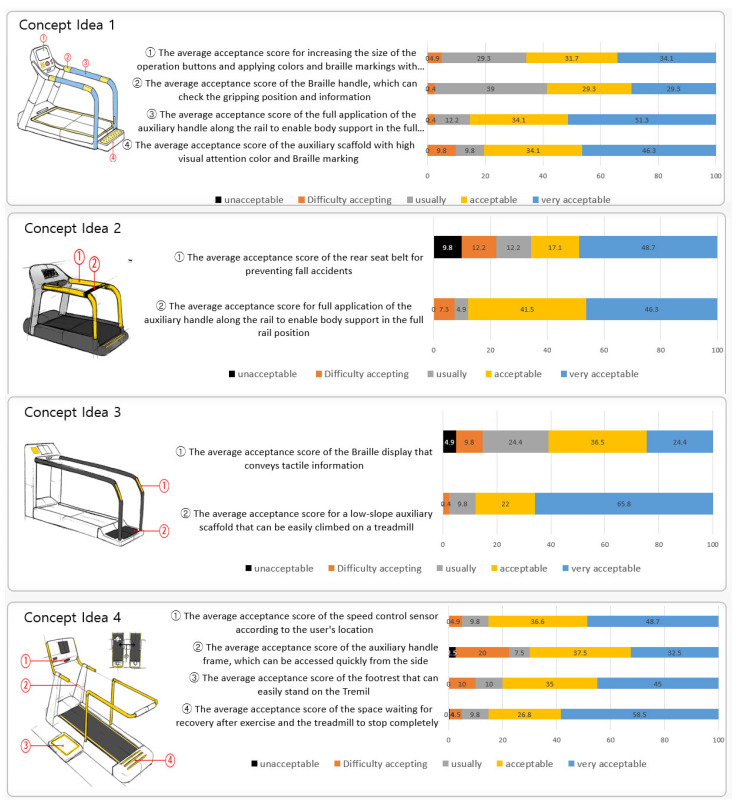
Evaluation result of concept idea acceptance of the treadmill with UD applied.

**Figure 8 healthcare-10-02278-f008:**
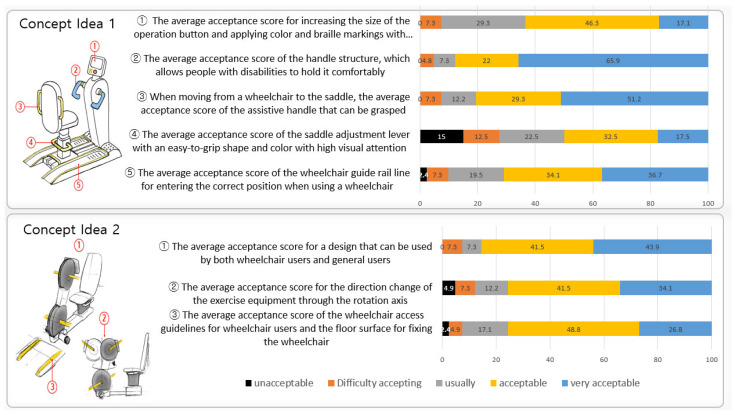
Concept idea of an arm ergometer applying UD.

**Figure 9 healthcare-10-02278-f009:**
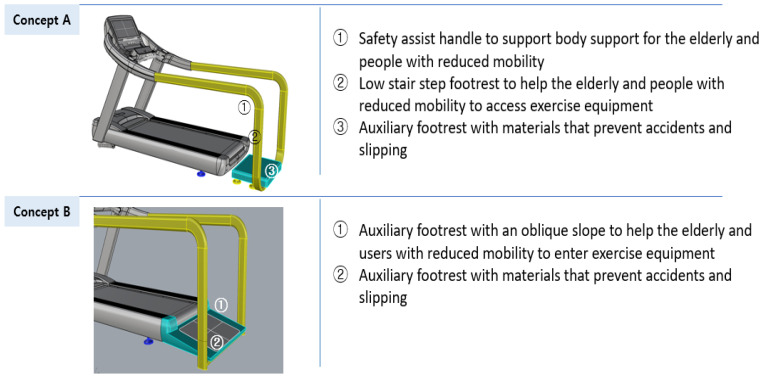
First 3D modeling of the treadmill design applying UD.

**Figure 10 healthcare-10-02278-f010:**
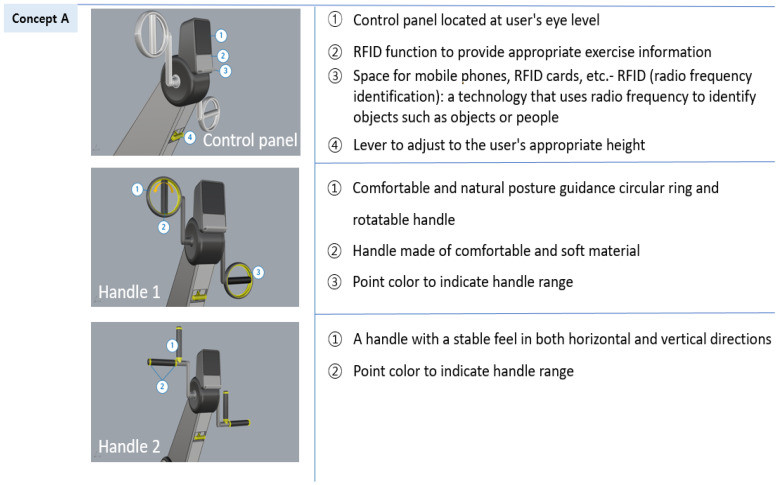
First 3D modeling of the arm ergometer design applying UD.

**Figure 11 healthcare-10-02278-f011:**
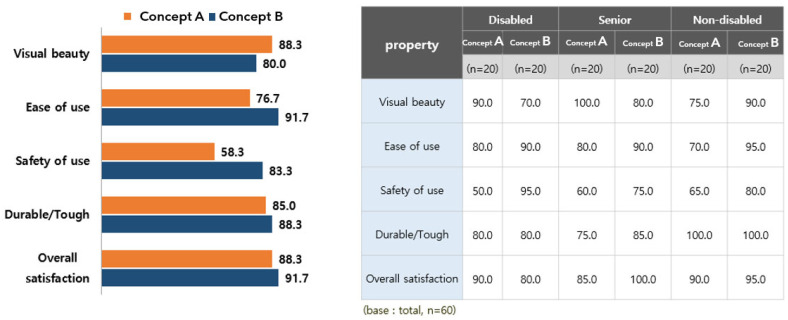
The result of attribute evaluation of the treadmill product design.

**Figure 12 healthcare-10-02278-f012:**
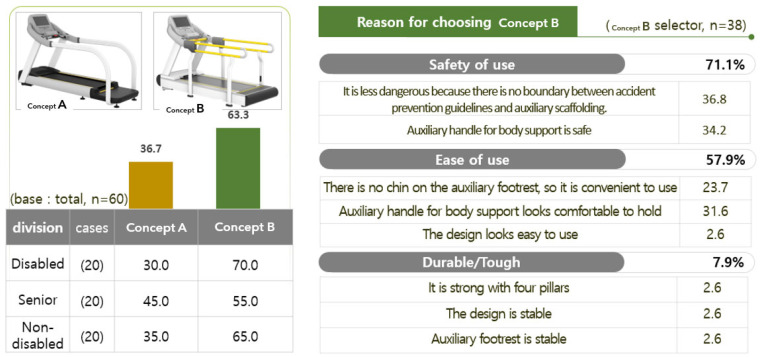
Results of the preference property evaluation of the treadmill product design.

**Figure 13 healthcare-10-02278-f013:**
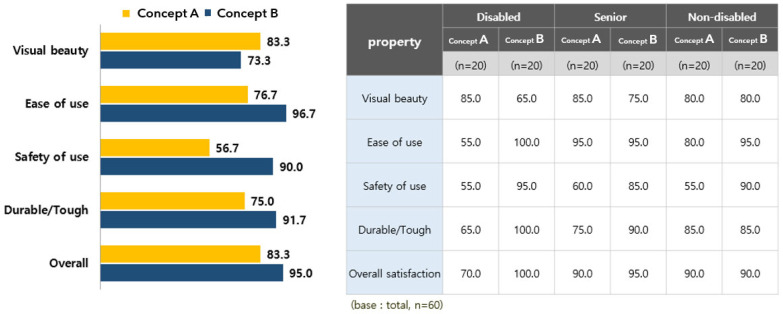
The result of the attribute evaluation of the arm ergometer product design.

**Figure 14 healthcare-10-02278-f014:**
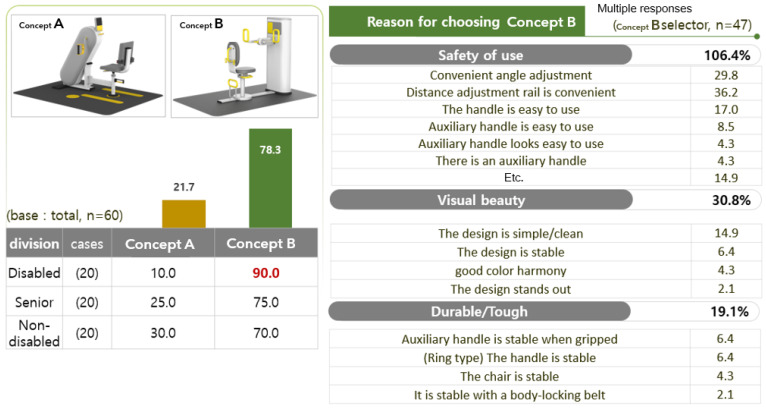
Results of the preference property evaluation of the arm ergometer product design.

**Figure 15 healthcare-10-02278-f015:**
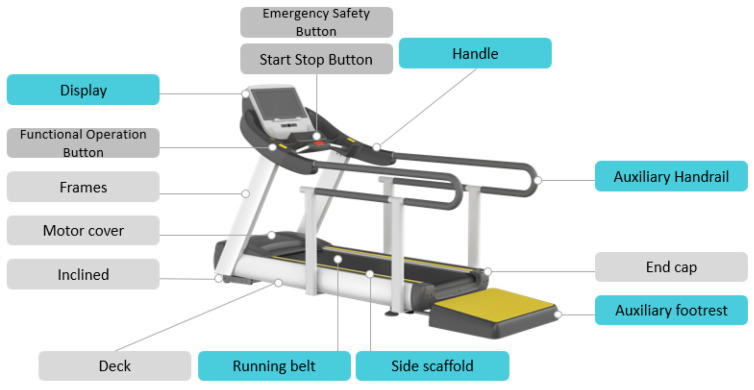
Second design concept of treadmill applying UD.

**Figure 16 healthcare-10-02278-f016:**
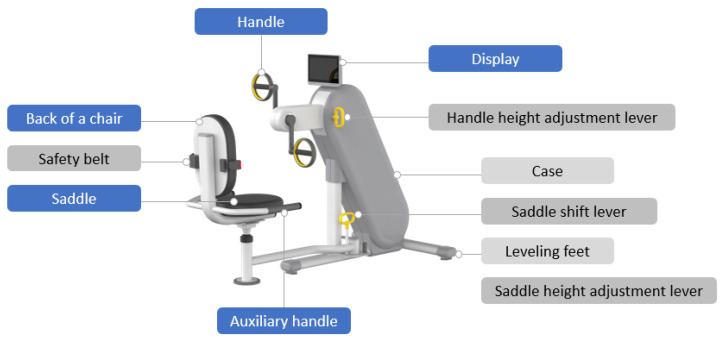
Second design concept of the arm ergometer applying UD.

**Figure 17 healthcare-10-02278-f017:**
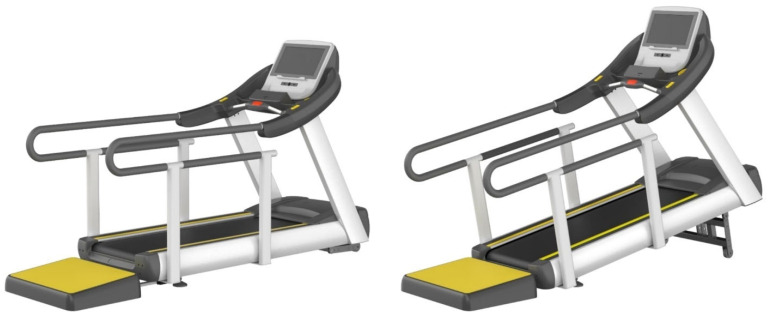
Main features of the treadmill design.

**Figure 18 healthcare-10-02278-f018:**
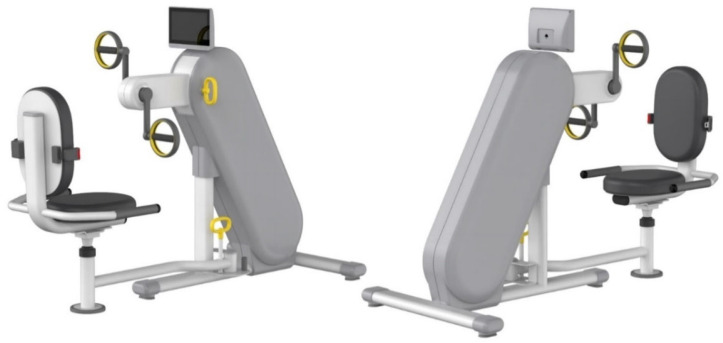
Main features of the arm ergometer.

**Table 1 healthcare-10-02278-t001:** Focus group interview participants.

Workplace	Region	Trainer Career (Year)
Seoul Underwater Rehabilitation Center	Seoul	22
Jeonglib hall	Seoul	11
Gwangju National Sports Center for the Disabled	Gwangju	13
Goyang Rehabilitation Sports Center	Gyeonggi	32
Seongnam Hanmaeum Welfare Center	Gyeonggi	9
Jecheon Eoullim Sports Center	Jecheon	6
Asan National Sports Center for the Disabled	Asan	11
Jeonju Ullim National Sports Center	Jeonju	11
Gwangyang National Sports Center for the Disabled	Gwangyang	8
Gumi City Disabled Gymnasium	Gumi	33
Changwon Municipal Gomduri National Sports Center	Changwon	6
Chuncheon Disabled Sports Center	ChunCheon	7
Seoul Gomduri Sports Center	Seoul	25

**Table 2 healthcare-10-02278-t002:** Expert meeting participants.

Workplace	Specialty	Position	Trainer Career (Year)
Hansung University	Design engineering	Professor	24
Saehan University	Design engineering	Professor	22
Hankyung University	Design engineering	Professor	24
Hongik University	Industrial design	Professor	28
Woosong University	Industrial design	Professor	25
Kookmin University	Industrial design	Professor	14
Seoul National University of Science and Technology	Design engineering	Professor	21
Seoul Design Foundation	Universal design	Senior Researcher	24
Korea University of Technology and Education	Design engineering	Professor	26
Korea Welfare University	Universal design	Professor	18

**Table 3 healthcare-10-02278-t003:** Characteristics of participants in the first design concept idea acceptance evaluation.

Division		People with Disabled (*n* = 133)		Senior (*n* = 131)		Non-Disabled (*n* = 132)		Total (*n* = 396)	
		*n*	%	*n*	%	*n*	%	*n*	%
gender	male	88	66.2	66	50.4	69	52.3	223	56.3
	female	45	33.8	65	49.6	63	47.7	173	43.7
age	20~29	12	9.1	0	0	18	13.6	30	7.6
	30~39	18	13.5	0	0	20	15.2	38	9.6
	40~49	22	16.5	0	0	33	25	55	13.9
	50~59	49	36.8	0	0	61	46.2	110	27.8
	60+	32	24.1	131	100	0	0	163	41.1
type of disability	Brain lesion	33	24.8	-					
	physical disability	86	64.7						
	Sensory impairment (visual, hearing)	14	11.1						
degree of disability	mild	76	57.1						
	severe	57	42.9						

**Table 4 healthcare-10-02278-t004:** Characteristics of 3D design satisfaction assessment participants.

Gender		Age					Participants		
Male	Female	20~29	30~39	40~49	50~59	60+	People with	Non-Disabled	Senior
							Disabled		
*n* (%)	*n* (%)	*n* (%)	*n* (%)	*n* (%)	*n* (%)	*n* (%)	*n* (%)	*n* (%)	*n* (%)
27 (45.0%)	33 (55.0%)	3 (5%)	7 (11.7%)	5 (8.3%)	15 (25.0%)	27 (45%)	10 (33.3%)	10 (33.3%)	10 (33.3%)

**Table 5 healthcare-10-02278-t005:** Aerobic exercise equipment used by both people with a disability and those with no disability.

England		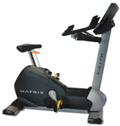	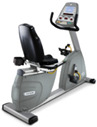	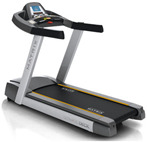
	Company	Matrix	Matrix	Matrix
	Product	Recumbent Cycle	Upright Cycle	Treadmill
	Standard	165.5 cm × 67 cm × 128 cm	130 cm × 65 cm × 153 cm	216 cm × 86 cm × 140 cm
	Target use	Blind and hearing-impaired users	Blind and hearing-impaired users	Blind and hearing-impaired users
	Certification	IFI Certification	IFI Certification	IFI Certification
USA		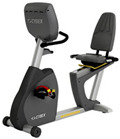	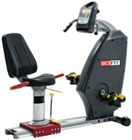	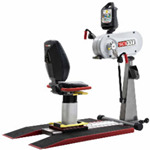
	Company	CYBEX	Sci-Fit	Sci-Fit
	Product	Recumbent Cycle	Recumbent Cycle	Upper Body Ergometer
	Standard	165 cm × 63.5 cm × 132 cm	150 cm × 66 cm × 132 cm	152 cm × 76 cm × 195 cm
	Target use	non-disabled, visually impaired, and elderly people users	Non-disabled, wheelchair users, disabled, elderly users (Removable Seat)	Non-disabled, wheelchair users, disabled, elderly users (Removable Seat)
	Certification	IFI Certification	IFI Certification	ADA Certification
Republic of Korea		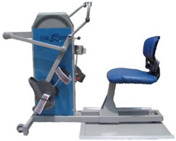	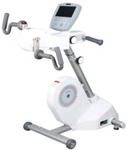	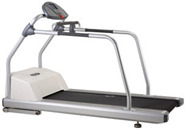
	Company	Shinkwang Industrial Co., Ltd.	Sungdo MC Co., Ltd.	Sungdo MC Co., Ltd.
	Product	Recumbent Cycle/Upper Body Ergometer	Upper Body Ergometer	Treadmill
	Standard	142 cm × 69 cm × 108 cm	55 cm × 92 cm × 108 cm	215 cm × 85 cm × 142 cm
	Target use	Non-disabled, wheelchair users, disabled, elderly users	Non-disabled, wheelchair users, disabled, elderly users	non-disabled, visually impaired, and elderly people users
	Certification	-	Korea Ministry of Food and Drug Safety Certification	Korea Ministry of Food and Drug Safety Certification

**Table 6 healthcare-10-02278-t006:** Aerobic exercise equipment FGI results of sports facilities.

Status	Most facilities do not have separate equipment for the disabled and the elderly (1 to 2 holding levels) Indoor exercise equipment has free-weight equipment-level exercise equipment 60–70% of users use aerobic exercise equipment, and among them, treadmills are highly used
Problem	In the case of treadmills, many users want to use them, but there are many concerns about safety issues. Safety guide bars, access to treadmill belts, etc. are commonly drawn from experts It is an aerometer that enhances the exercise effect of both the elderly and the disabled with aerobic exercise equipment that they hope to be equipped with, and no product that can be used by the disabled and non-disabled in compatibility
Needs	Requires separate equipment for the disabled or a form of equipment that can be used simultaneously by both disabled and non-disabled Most facility managers prefer equipment that allows disabled and non-disabled people to share a single product Aerobic exercise equipment requires treadmills and upper limb ergometers, and due to space problems in the playground, exercise equipment with UD for both disabled and non-disabled people is required

**Table 7 healthcare-10-02278-t007:** Problems of existing treadmills and concept ideas applying universal design.

Problems and needs	Access	Difficulty climbing the rails for the blind and the elderly Difficulty in recognizing movement information for visually impaired people Difficulty recognizing that blind people are standing in the proper position on the rails	
	Execution Function Control	It is difficult for the blind and the elderly to operate the button Inability to perceive movement status information for visually impaired people Control of additional function buttons such as speed, slope, and time is confusing Difficulty operating the speed control button when walking or running at high speed A sharp increase in speed can result in loss of center and injury Risk of accident if walking speed is lower than rail speed Need something to support when on the back of the rail	
	End	It is difficult for the visually impaired to recognize the degree of deceleration of the rail Blind and elderly people may be injured when descending from the apparatus Injury may result from coming down from the rail before the rail has come to a complete stop	
Concept Idea		Increase the size of the operation button and apply color and braille notation with high visual attention Introduction of voice guidance system Introduced braille markings on the auxiliary handle at the end of the rail Auxiliary footrest with high visual attention color and Braille markings High visual attention color applied to the rail side part Automatic speed adjustment function according to the rail position where the user is standing Shock-absorbing material applied to auxiliary handle to minimize the risk of injury Fully applied auxiliary handle along the rail for body support in full rail position Automatic emergency stop function if the user is positioned at the end of the rail during exercise	

**Table 8 healthcare-10-02278-t008:** Problems of existing arm ergometer and concept ideas applying universal design.

Problems and needs	Access	Difficulty in recognizing the manual for the visually impaired and the elderly Difficulty entering the proper location for wheelchair users when using a wheelchair When a wheelchair user uses a wheelchair, it is difficult to set an appropriate distance between the handle and the wheelchair. Inconvenient for wheelchair users to move chairs when using a wheelchair Inconvenient for wheelchair users to move into a chair when using a chair
	Execution Function Control	Difficulty recognizing the position of the instrument handle and adjustment knob for the visually impaired When a wheelchair user uses a wheelchair, effective upper-body movement is difficult It is difficult for wheelchair users to set the appropriate height when using a wheelchair When a wheelchair user uses a wheelchair, a stable wheelchair fixation is required It is difficult for beginners to recognize the right chair height for them Difficulty in adjusting the chair distance Difficulty in adjusting the height of the chair Users with limited hands may find it difficult to hold the handle and apply continuous force It is difficult for users with upper body discomfort to fix the upper body to the backrest It is difficult for users with lower limbs to fix their feet on the floor
	End	Inconvenient to transfer to a wheelchair if a wheelchair user uses a chair
Concept Idea		Handle structure for a comfortable grip even for people with disabilities Handle fastening belt to secure your hand to the handle Increase the size of the operation button and apply color and braille notation with high visual attention Handle with high visual attention color applied The tilting structure of the instrument station considering the accessibility of both wheelchair users and non-users Devise a saddle tilting method that can be easily adjusted by one person Guide rail line for entry into the correct position when using a wheelchair Auxiliary handle to hold when moving from the wheelchair to the saddle Inventing an easy and intuitive way to adjust the saddle distance Saddle adjustment lever design with an easy-to-grip shape and high-visibility color

**Table 9 healthcare-10-02278-t009:** Treadmill guidelines with UD applied.

Running belt	Provides automatic speed control to prevent falls Easy access to anyone when using treadmill Use highlights to display guidelines for belt and side platform separation Use durable material on the belt
Auxiliary footrest	Provide an auxiliary footrest to aid in access to the running belt A non-slip material should be used for the auxiliary footrest Use contrasting colors to mark guidelines to distinguish them from running belts
Handles and Auxiliary Handrails	Provide auxiliary handrails for exercise assistance and accident prevention The handle and auxiliary handrail should be of a shape and size that is easy for anyone to hold Handles and auxiliary handrails must be made of non-slip materials The height of the handle and auxiliary handrail should be positioned in consideration of the body dimensions of various users Handles and auxiliary handrails should not interfere with the user’s movement and range of motion
Display	Information on the screen should be displayed easily and clearly Use a readable size for the text on the screen The text on the screen should use a color considering the clarity Provide appropriate feedback so that information such as operation status and operation results can be easily recognized Provide information in a variety of ways, such as visual + auditory, visual + tactile, etc.
Operation button	The operation buttons should be easy and intuitive to use Display the name of the operation button in text and braille The operation button should be easy to press with little force The operation button must be of a size and shape that can be operated accurately by all users The height and position of the operation button should be placed in consideration of the body dimensions of various users The operation buttons should be placed in consideration of the frequency of use The emergency safety button should be placed in an easily visible location The emergency safety button should be marked with a pictogram and highlighted color
Etc.	Provide a sufficient range of safety space around the instrument

**Table 10 healthcare-10-02278-t010:** Arm ergometer guidelines with UD applied.

Saddle	It must be in the form of a backrest to support the user’s body Provide auxiliary seat belts for users with limited body support Provide saddle-assist handles for users with reduced mobility It should be easy for wheelchair users to move when using the saddle The saddle must be movable for wheelchair access Provide guidelines guiding the correct location when entering a wheelchair The operation of the saddle adjustment lever should be easy and intuitive Provide braille markings on saddle adjustment levers Use an accent color for intuitive recognition of the saddle adjustment lever. Use a soft, durable material for the saddle
Handle	The handle should be of a shape and size that is easy for anyone to hold Handles should be made of non-slip material The handle should be placed in consideration of the body dimensions of various users, and its position and height should be adjustable The handle should not interfere with the user’s movement and range of motion
Case	Cases and frames must not interfere with the user’s movement and range of motion The case and frame must not impede access to the wheelchair
Display	Information on the screen should be displayed easily and clearly Use a readable size for the text on the screen The text on the screen should use a color considering the clarity. Provide information in a variety of ways, such as visual + auditory, visual + tactile, etc. The operation of the screen and buttons should be easy and intuitive Provide appropriate feedback so that information such as operating status and results can be easily recognized
Information board	Guide information and cautions so that anyone can easily recognize them Information boards should be placed in consideration of the views of various users Include images or pictograms of information boards Provide information on information boards in Braille Text on information boards should be readable in size The text and background color of the information board should use a color considering the clarity
Etc.	Provide a sufficient range of safety space around the instrument The floor around the device should be made of non-slip material to support the feet

## Data Availability

The data presented in this study are available on request from the author.

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
