# Peer review of "Development of Aerobic Exercise Equipment Using Universal Design: Treadmill and Arm Ergometer"

_healthcare, 2022, doi:10.3390/healthcare10112278_

Round 1
Reviewer 1 Report
The paper must be improved in order to be published in the next terms:
- The introduction in page 2 is correct but it should be separated into different paragrath in order to be more readable
- The last part of the sentence in line 113 makes no sense
- The letter type of reference 21 in line 113 is different
- Try to avoid big blank spaces at the end of the pages (3, 4, 5, 8, ...)
- In Table 1 Gyeonggi must start with capital letter
- In Table 1 the column Triner is disaligned
- In line 142 "was derived" must be deleted
- In line 147 revise "is designed is evaluated"
- The type of letter of the paragrath from line 153 to 158 must be changed
- In Table 4, it should be explained what is each of the 5 files
- In line 176 change the reference to the Table as it is incorrect (Table 00)
- In section 3.2.1, Figures 3 and 4 and Tables 6 and 7 must be explained
- In Tables 5, 6, 7, 8 and 9 it is better to justify the text to the left instead of centering it as it would be more readable
- Some figures (3, 10, ...) have some text underlined as they are images captured and some word are marked as not recognized by the word processor
- Figure 4 is too small and the text is difficult to read
- References to Figure 11 and the following have the characters '<' and '>' (<Figure 11>). Delete those characters.
- Figure 11 shows 'A type' and 'B type' but they have not been defined. Are they 'Concept A' and 'Concept B'? If so, change them.
- Sections 3.4.1 and 3.4.2 must be more explained in text as it has almost only two figures and tables without any type of explanation
- The first sentence of section 4 (lines 287 and 288) makes no sense. Rewrite it.
- When referencing articles with more than two authors use et. al instead of writing all the surnames (specially in section 4)
- The paper proposes some improvements over two types of exercise machines. How much would be the overcost of those improvements?
- Section 6 is named Patents. I think this is not the correct name
Author Response
첨부파일을 참조하시기 바랍니다.

Reviewer 2 Report
The current work showed a good aim to the current needs for the inclusion of disabled people into a more physically active lifestyle. Nevertheless, some observations need to be addressed before I made a final judgement:
Corrections
Ln 21 “…Future research is as follows.”
Ln 35-37 Please, re-structure the sentence to a clearer one, also what is the source of the survey?
Ln 39 Delete “is” from the higher “is” the demand
Ln 40 replace the coordinating conjunction “for” with other options, seems confusing the sentence.
Ln 42-43 Please enlist additional disadvantages from the disabled community
Ln 51-53 Please, add the sources of the psychological implications and social interaction from your target population
Ln 57 The term exclusive seems like it’s focused on the population, please reword.
Methods.- Was there an specific statistical analysis besides descriptive to the satisfaction survey? What about the equal amount or proportion comparison among non-disable and disabled people?
A more intensive analysis and sample assessment among surveys must be displayed in the methods section.
Figure 8. Are the bar numbers percentiles of the assessed population? Please indicate the final total sample size number.
Figure 11. Why the concept design contrast has lower sample interview? Please specify in figure foot.
Reviewer 3 Report
General comments:
This study focuses on the designing exercise equipment for the disabled. Authors explored the components of universal design and product satisfaction by considering users with and without disabilities and proposed the direction for designing extended universal disabled exercise equipment.
The study surprised me positively as it focused on an interesting and relevant issue which is adapting existing equipment so that disabled people can have easy access to physical activity.
The introduction and the methodology are appropriate and well written. The final results show the importance of starting to produce equipment aimed at non-disabled and disabled people.
I wonder if it would be interesting to include in the discussion some reference to cost-benefit analysis.
After reading the document I did not clearly understand if your results can be used and are open to any exercise equipment manufacturers.
I would recommend a general revision because several minor mistakes were found while reading the article.
Round 2
Reviewer 1 Report
- Figure 6 presents the data using an incorrect order (4, 5, 6, 1, 2, 3,). Reorder
- Figures 7 and 8 has underline text ("Concept Idea X") as they are captured images from a word editor
- References are made using the "[number]" format. However, in section 4 the format changes using the format "author's surnames [number". Formats must be unified
- In section "Author Contributions" delete the text "For research ... should be used"
